# Incorporating contextual information into KGWAS for interpretable GWAS discovery

**Cheng Jiang**[1][§] **Brady Ryan**[1][§] **Megan Crow**[2] **Kipper Fletez-Brant**[2]
**Kashish Doshi**[2] **Sandra Melo Carlos**[2] **Kexin Huang**[3]
**Heming Yao**[2][†] **Burkhard Hoeckendorf**[2][†] **David Richmond**[2][†]

[1] University of Michigan    [2] gRED, Genentech    [3] Stanford University
[§]Work conducted during an internship at Genentech
[†]{yao.heming, hoeckendorf.burkhard, richmond.david}@gene.com

## Abstract

Genome-Wide Association Studies (GWAS) identify associations between genetic variants and disease; however, moving beyond associations to causal mechanisms is critical for therapeutic target prioritization. The recently proposed Knowledge Graph GWAS (KGWAS) framework addresses this challenge by linking genetic variants to downstream gene-gene interactions via a knowledge graph (KG), thereby improving detection power and providing mechanistic insights. However, the original KGWAS implementation relies on a large general-purpose KG, which can introduce spurious correlations. We hypothesize that cell-type specific KGs from disease-relevant cell types will better support disease mechanism discovery. Here, we show that the general-purpose KG in KGWAS can be substantially pruned with no loss of statistical power on downstream tasks, and that performance further improves by incorporating gene–gene relationships derived from Perturb-seq data. Importantly, using a sparse, context-specific KG from direct Perturb-seq evidence yields more consistent and biologically robust disease-critical networks.

## 1 Introduction

Recent studies estimate that drug discovery programs with genetically supported targets are 2–3 times more likely to succeed (King et al., 2019). Genome-wide association studies (GWAS) have become a cornerstone of this effort, mapping thousands of variant–trait associations. However, GWAS signals alone don't reveal the causal variants, relevant cell types, or regulatory mechanisms driving disease initiation and progression. Consequently, there is an urgent need for methods that can translate statistical associations from GWAS into actionable mechanistic insights.

Bridging the gap between genetic variants and disease mechanisms requires integrating diverse and often disconnected data sources. Traditional approaches map genetic variants to candidate genes using functional genomics (Li & Ritchie, 2021). However, variant-to-gene mapping often fails to elucidate specific disease mechanisms, as genes frequently participate in multiple biological pathways whose relevance varies across cellular contexts. To address these complexities, recent efforts have turned to Knowledge Graphs (KGs). A prominent example is Knowledge Graph GWAS (KGWAS), a framework that links genetic variants to biological programs end-to-end using geometric deep learning (Huang et al., 2024). By leveraging prior biological knowledge, KGWAS increases statistical power to detect disease-associated variants, particularly in small cohorts, and derives interpretable disease-critical networks via graph attention.

Despite its promise, the standard KGWAS framework relies on a large, general-purpose KG constructed from broad data sources to ensure applicability across diverse traits. In contrast, GWAS signals are usually sparse: while millions of variants are tested, only hundreds to thousands reach significance for a given trait (Yengo et al., 2018). We hypothesize that models built on such large, generic KGs may be sensitive to spurious correlations and lack the resolution required for specific disease contexts.

Motivated by the goal of understanding disease mechanism, we propose to trade generality for improved contextual relevance (Figure 1). We focus on a set of closely related traits and explore

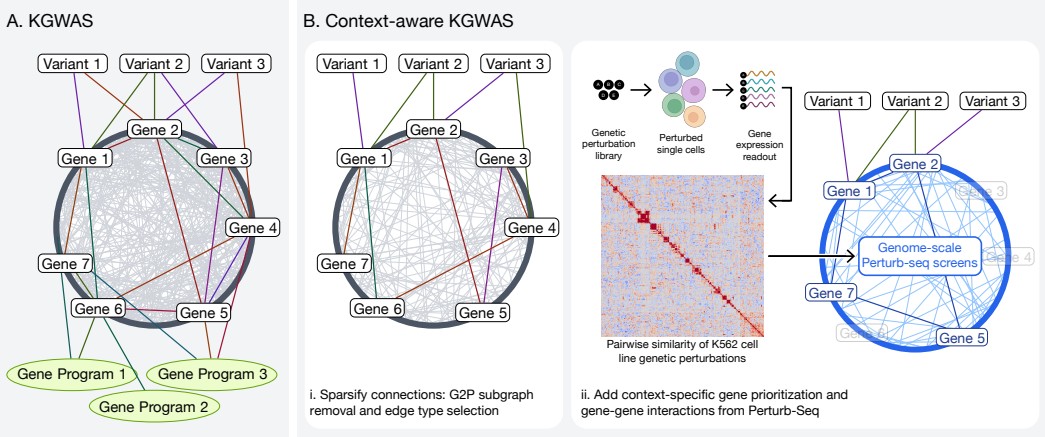

Figure 1: Knowledge graph construction. A. The original KG in KGWAS consisting of variant-gene, gene-gene and gene-program edges; B. Our extension to KGWAS: (left) removing gene-program edges, and sparsifying remaining connections; (right) replacing gene-gene edges with contextually relevant relationships derived from Perturb-seq.

strategies to prune the general-purpose KGWAS knowledge graph, making it more trait-specific by incorporating direct experimental evidence from Perturb-seq. We demonstrate that the KG can be substantially pruned without degrading performance, and that integrating Perturb-seq data greatly improves significant loci retrieval in small cohorts. The resulting model, which we refer to as context-aware KGWAS, yields more consistent disease-critical networks, providing deeper, actionable insights into the mechanisms underlying complex traits.

## 2 RELATED WORKS

The increasing availability of large-scale perturb-seq datasets, has enabled the recent development of methods for the mechanistic interpretation of GWAS. One such method, known as V2G2P (Schnitzler et al., 2024), integrates multiple V2G methods (Nasser et al., 2021; Fulco et al., 2019; Karolchik et al., 2011) and Perturb-seq to map variants to genes and genes to biological programs in relevant cell type context. Leveraging this framework, the authors identified multiple coronary artery disease GWAS signals that converge onto transcriptional programs related to the cerebral cavernous malformations signaling pathway, demonstrating the value of using cell-type-specific experimental data to delineate the path from genetic variants to cellular mechanism. Building on the utility of perturbation data, Ota et al. (2025) introduced a causal modeling approach that combines loss-of-function (LoF) burden tests with Perturb-seq. Unlike standard GWAS, LoF burden tests provide quantitative and directional estimates of gene-trait relationships. By averaging these effects across genes within programs identified from Perturb-seq, the authors constructed causal graphs linking regulators to biological programs and, ultimately, to traits such as mean corpuscular hemoglobin (MCH).

In contrast to the above methods, which rely on *de novo* program discovery from perturbation screens in specific cell lines, Knowledge Graph GWAS (KGWAS; Huang et al., 2024) leverages a massive functional genomics knowledge graph (KG) containing millions of interactions, including variant-to-gene (V2G), gene-to-gene (G2G), and gene-to-program (G2P) links from curated databases. KGWAS trains disease-specific graph attention networks to predict disease associations, using the KG as a functional prior to significantly improve detection power in small-cohort GWAS. Additionally, KGWAS also provides insight into "disease critical networks" through attention-based interpretability.

Our proposed method builds on the observation that cell-type-specific functional genomics data is critical for understanding the mechanism underlying GWAS. Specifically, we leverage the KGWAS framework, incorporating Perturb-seq data directly into the knowledge graph structure. By combining the predictive power of geometric deep learning with contextually relevant perturbation data, we aim to improve both the discovery of novel variants in data-scarce regimes and the mechanistic interpretation of their underlying molecular networks.

## 3 METHODS

We adopt and extend the KG and heterogeneous graph attention network architecture introduced by KGWAS (Huang et al., 2024). We begin by briefly describing the graph structure and underlying message-passing scheme, followed by our adaptations to sparsify the KG and incorporate context-aware functional genomics information from Perturb-seq. Diagrams comparing KG construction in KGWAS and our resulting model are shown in Fig. 1.

### 3.1 KNOWLEDGE GRAPH FORMULATION

The functional genomics knowledge graph is defined as $\mathcal{G} = (\mathcal{V}, \mathcal{E}, \mathcal{T}_R)$, where $\mathcal{V}$ denotes the set of nodes, $\mathcal{E}$ the set of edges, and $\mathcal{T}_R$ the set of relation types. The nodes $\mathcal{V}$ are partitioned into three distinct types: genetic variants (SNPs), genes, and biological programs. The KG architecture contains 784,708 variants from the UK Biobank (Bycroft et al., 2018), 23,746 protein-coding genes, and 28,714 gene programs from the Gene Ontology resource (GO; Ashburner et al., 2000). The set of relations $\mathcal{T}_R$ preserves the biological source of each interaction, categorized into three primary types:

- **Variant-to-Gene (V2G):** Encodes functional links via 14 edge types (e.g., eQTL, ABC, PCHi-C), totaling 8.63 million edges.
- **Gene-to-Gene (G2G):** Captures 2.33 million edges across 46 relation types, including protein-protein interactions and signaling pathways from BioGRID and STRING.
- **Gene-to-Program (G2P):** Maps 116,610 edges via 10 relation types based on GO annotations and biological pathway memberships.

Node embeddings are initialized as follows: variants are initialized using 70 functional annotations, including chromatin accessibility, conservation scores, and allele frequencies; genes are initialized with PoPS features (Weeks et al., 2023); programs are randomly initialized.

### 3.2 HETEROGENEOUS GRAPH ATTENTION NETWORK

For each trait, a heterogeneous graph attention network (GAN) is trained to predict GWAS $\chi^2$ association statistics from the learned variant embeddings. The training process utilizes an LD-aware loss function to account for the correlation between variants in linkage disequilibrium (Huang et al., 2024). The resulting predicted $\chi^2$ statistics are then used to generate a new set of association p-values by weighting the original GWAS p-values while controlling the false discovery rate (Genovese et al., 2006). This architecture allows the GAN to perform inductive reasoning over the functional KG, prioritizing variants whose biological context suggests a higher probability of true disease association.

### 3.3 KNOWLEDGE GRAPH SPARSIFICATION

While KGWAS achieves significant improvements over standard GWAS in small cohorts, the constructed KG is very large and contains substantial redundancy. Such redundancy can generate multiple plausible paths to represent the same underlying biological mechanism, ultimately reducing the generalization, interpretability and consistency of disease-critical networks. To address this issue, we conduct the following ablation studies to evaluate the contribution of specific edge types:

**Disentangling G2G and G2P Contributions:** Both relations connect genes sharing similar biological functions — G2G through direct interaction and G2P through pathway membership. We independently evaluate sub-graphs $\mathcal{G}_{G2G}$ and $\mathcal{G}_{G2P}$ to determine if physical interaction data and pathway annotations provide complementary or redundant signals.

**Edge Type Selection:** We hypothesize that low-specificity edges may dilute the message-passing signal. For example, a V2G edge type that links variants to genes solely by proximity to a transcription start site (TSS) is very broad, connecting up to the closest 20 genes (Gazal et al., 2022), and thus includes associations that lack high-confidence functional evidence, yet it accounts for 67% of all V2G edges. We define a high-confidence V2G subset, $\mathcal{T}_{V2G}^{local}$, restricted to cis-regulatory relationships with strong functional evidence (exon, promoter, eQTL, and fine-mapped eQTLGen). Similarly, we

observe an additional 1.09 million G2G self-loop edges, arising from the model construction that adds self-loops for every combination of gene and edge type. Removing sparsely connected edge types eliminates relations dominated by redundant self-loops, reducing unnecessary parameterization and preventing dilution of informative gene-gene signals. We define $\mathcal{T}_{G2G}^{major}$, which retains only edge types with $> 10,000$ connections in the graph, thereby reducing the total number of edge types from 46 to 12. The retained edge types are: physical association, reaction, catalysis, binding, literature, signaling, complexes, activation, binary, inhibition, kinase, and metabolic. Additionally, we evaluate the effect of collapsing all V2G edges to a single type, and likewise with G2G edges.

### 3.4 CONTEXT-AWARE KNOWLEDGE GRAPH FORMULATION

The baseline KGWAS graph reflects universal biology, aggregating relationships across diverse cell types. However, gene regulation is inherently context-dependent. For example, single-cell ATAC-seq atlases show that many cis-regulatory elements are differentially accessible across cell types and cellular states (Zhang et al., 2021). To enhance the model's specificity for disease traits, we incorporate cell-type-specific interaction data derived from Perturb-seq screens (Dixit et al., 2016).

We construct a context-specific G2G relation type based on the pairwise cosine similarity of transcriptional responses, such that genes inducing a similar transcriptional response upon perturbation are connected by an edge. Our underlying assumption is that similar perturbation response reflects functional associations of the perturbed genes. Context-specific G2G edges $\mathcal{E}_{G2G}^{Perturb}$ are derived by binarizing the absolute cosine similarity scores between transcriptional responses using a threshold $\tau$. By integrating these cell-type-specific edges, the model can prioritize pathways that are functionally active in the disease-relevant cellular context.

## 4 EXPERIMENTAL DESIGN

### 4.1 PRE-PROCESSING OF PERTURB-SEQ DATA

Perturb-seq is a pooled CRISPR-based genetic screen that combines targeted perturbations with single-cell RNA-seq (Dixit et al., 2016). These experiments systematically perturb (knock down or activate) genes and measure the resulting transcriptional responses in individual cells. We utilize data from an experimental genome-scale Perturb-seq screen targeting all essential and expressed genes in the human leukemia-derived cell line K562 using CRISPR interference (Replogle et al., 2022).

We apply standard Perturb-seq processing to remove doublets based on guide RNA assignments (Meyers et al., 2023). For each perturbation, we compute log-fold changes (LFC) for each feature gene using non-targeting controls as the background (Dixit et al., 2016). We then z-score the LFC matrix and select 5,000 features comprising 3,000 feature genes with high binomial deviance and 2,000 highly variable feature genes. Finally, we apply independent component analysis (ICA; Lee, 1998) for dimensionality reduction with 60 components, and compute the cosine-similarity between program activations for each pair of gene targets.

To assess the biological signal in the K562 Perturb-seq dataset, we compare the pairwise cosine similarities for gene targets with known relationships documented in STRING (Szklarczyk et al., 2021) (using high confidence cutoff) relative to randomly selected gene pairs (Fig. 2). Edges corresponding to STRING annotations have significantly higher cosine similarities, confirming that the Perturb-seq phenotype embeddings capture meaningful biological information. Based on this result, we set $\tau = 0.5$ for $\mathcal{E}_{G2G}^{Perturb}$, yielding 116k high-confidence context-specific G2G edges and minimizing inclusion of lower-confidence relationships.

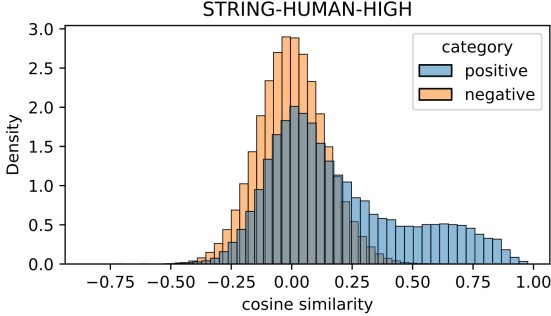

Figure 2: Distribution of cosine similarities between pairs of target genes annotated in STRING (positive) and randomly selected pairs of target genes (negative).

## 4.2 Model Implementation and Training

Our implementation of the full framework and the training protocol follows the KGWAS repository[1]. For each trait, the model is trained for up to 10 epochs using the Adam optimizer, with an early-stopping strategy based on validation mean squared error using 5% of the variants. For each combination of trait and sample size, we train three models with different random seeds.

## 4.3 Evaluation Criteria

We also adopt the evaluation pipeline from KGWAS. For each trait, we define the ground-truth associations as the 100 most significant loci from GWAS performed on the full UK Biobank cohort (Sudlow et al., 2015). We evaluate the ability of KGWAS to correctly identify ground truth associations from sub-sampled cohorts. Each cohort is subsampled to include 1000, 2500, 5000, 7500, 10000, and 50000 individuals. We report the number of loci in the top 100 independent risk loci identified by each method that overlap with loci discovered in the full cohort.

We focus on three traits: Mean Corpuscular Hemoglobin (MCH), Immature Reticulocyte Fraction (IRF), and Red Cell Distribution Width (RDW). These traits were previously selected by Ota et al. (2025) because open chromatin regions in K562 exhibit significant heritability enrichment for these traits, consistent with the erythroid lineage of K562. To obtain the ground truth association from the full cohort and $\chi^2$ association statistics used for model training at each subsample cohort size, we perform GWAS using PLINK2.0 (Purcell et al., 2007) for MCH and IRF. For RDW, we directly use the GWAS results released by Huang et al. (2024).

## 4.4 Interpretability

After training the graph attention network model, we follow the approach introduced in KGWAS (Huang et al., 2024) to derive disease-critical networks. Specifically, we calculate the pre-softmax attention score for every edge. This score reflects the edge-level contribution from the neighborhood in predicting the $\chi^2$ statistics, which indicate the association with the disease. In the presence of multiple edge types, the attention scores of each edge type are normalized and then aggregated for the same pair of nodes across edge types using max-pooling. For each variant, we select the $k$ genes with the highest attention scores, and for each of those genes, we further select the $k$ genes to which they assign the highest attention. In this way, the most important V2G and G2G edges are included in the disease-critical network.

# 5 Results

## 5.1 Knowledge Graph sparsification without loss of performance

We evaluate the contribution of different node types, edge types, and information sources on the performance of GWAS hit detection across three traits (MCH, IRF, RDW). In this section, we report results for a small cohort sample size of 10,000.

### 5.1.1 Ablation of G2G and G2P edges

We ablated G2G and G2P edges, individually and in combination, and measured KGWAS performance (Fig. 3A). The number of G2P edges are far fewer than the number of G2G and V2G edges, and removing them does not significantly impact performance, suggesting that gene programs are not currently playing a large role in KGWAS. By contrast, G2G edges constitute up to 26% of all KG edges, and removing them substantially degrades performance.

We hypothesized that the limited contribution of G2P edges to model performance is due to insufficient depth of the two-layer GAN, prohibiting information flow between variants via connected genes and shared gene programs, which would necessitate three hops in the graph. To test this, we increased the GAN depth from two to three layers and re-evaluated models with G2G edges pruned. As shown in the second half of Fig. 3A, the three-layer models without G2G edges match the performance of

---

[1] https://github.com/snap-stanford/KGWAS

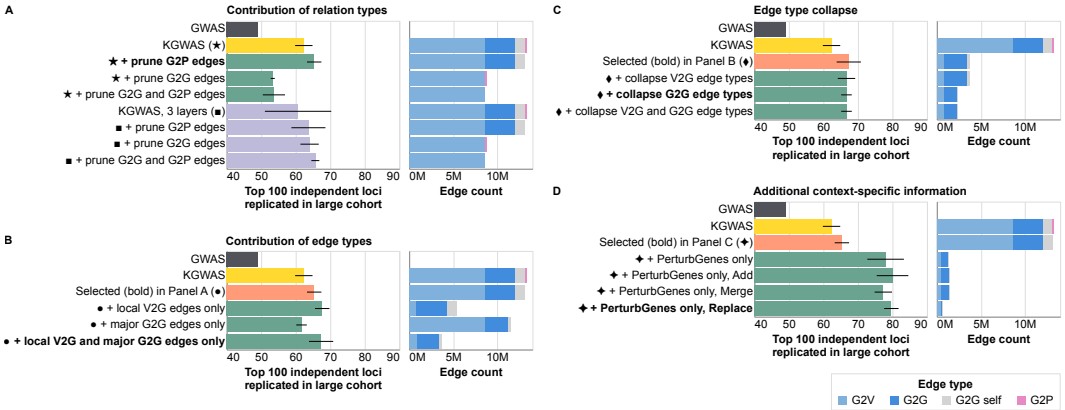

Figure 3: Contributions of different nodes and edge types in the KGWAS knowledge graph using a sample size of 10,000. Reported metrics are the total number of recalled independent loci summed across three selected traits. Standard deviations are computed across three training runs with different random seeds. In each subplot, bold indicates the model selected as the baseline in the subsequent panel. (A) Ablation of different relation types; (B) Ablation of V2G and G2G edge types; (C) Collapsing V2G and G2G edge types; (D) Adding context-specific information.

models with G2G edges. Additionally, in the three-layer GAN, KGWAS performance is resilient to removing both G2G and G2P edges. This underscores the redundancy within the densely connected knowledge graph. When direct G2G edges are absent, additional model depth can recover gene-gene communication through alternative paths, such as V2G edges. Based on these observations, for the remainder of this study we leverage a two-layer GAN, as in Huang et al. (2024), and remove G2P edges (equivalent to pruning program nodes) to minimize redundant paths and simplify interpretation of inferred disease critical networks.

### 5.1.2 Ablation of V2G and G2G edge types

We observed that some edge types, such as "20 closest to the TSS", have low specificity and may dilute higher-confidence edge types. In addition, edge types such as pQTLs include trans-regulatory effects that may be redundant with gene-to-gene (G2G) connections in the KG. Thus, we restricted V2G edges to cis-regulatory edge types within a local window (local V2G edges, $\mathcal{T}_{\text{V2G}}^{\text{local}}$), reducing the number of V2G edges by 10-fold. In addition, we noticed that due to the graph construction, which adds self-edges to every gene node for every edge type, G2G edge types with few connections contain a disproportionately high fraction of self-edges. To address this, we retained only G2G edge types with more than 10,000 connections in the KG (major G2G edges, $\mathcal{T}_{\text{G2G}}^{\text{major}}$), reducing the number of the G2G self-edges by 3.5-fold. As shown in Fig. 3B, restricting to local V2G edges slightly improves performance, and removing minor G2G edge types does not harm performance when combined with local V2G edges.

Finally, we evaluated collapsing multiple edge types to a single type, simplifying the heterogeneous knowledge graph used by KGWAS (Fig. 3C). Collapsing V2G edge types does not appreciably change the total edge count or improve performance – most of the V2G reduction comes from pruning to local edges. In contrast, even after pruning, G2G edges remain redundant, and collapsing G2G edge types further reduces edges without degrading performance. Based on these results, we use only local V2G edges, and major G2G edges collapsed to a single type for the remainder of this study.

### 5.1.3 Addition of context-specific genes and edges

We introduce contextual information by prioritizing genes that are significant for the K562 cell line, and should therefore be relevant for our selected traits related to hematopoiesis (MCH, IRF, RDW). Specifically, we restrict KGWAS gene nodes to the set of genes (n = 9,831) perturbed in Replogle et al. (2022), which include universally essential genes as well as genes measurably expressed in K562 cells. Building the graph using only these genes further reduces edge count, while simultaneously yielding a performance gain of over 10% (Fig. 3D).

Next, we evaluated the effect of context-specific G2G edges by augmenting or replacing the original G2G edges with edges that were inferred from Perturb-seq (see Section 4.1). We introduced context-specific G2G edges via three different strategies: (1) Add: addition as a novel edge type (heterogeneous graph), (2) Merge: merging with existing G2G edge type (simple graph), and (3) Replace: removing the original G2G edges and replacing with context-specific edges. Context-specific G2G edges are much sparser than the original G2G edges, therefore adding or merging increases the total G2G edge count only slightly, whereas replacing the original G2G edges results in a notable decrease in edge count. As shown in Fig. 3D, the three strategies perform similarly, indicating that a much smaller, context-specific G2G layer can substitute for a denser, heterogeneous G2G layer without loss of accuracy. For the final context-aware KGWAS model, we use the replace strategy, achieving a reduction of 19-fold in the total number of edges, while increasing performance by 20% over the original KGWAS model.

## 5.2 CONTEXT-AWARE KGWAS IMPROVES HIT DETECTION OVER A RANGE OF COHORT SIZES

Based on the observations in Sec. 5.1, we construct the final context-aware KGWAS graph by combining multiple strategies: (1) removing all gene–program nodes, (2) removing a subset of V2G and G2G edge types, (3) collapsing remaining G2G edges to a single type, (4) removing non-essential and non-expressed genes, and (5) restricting G2G edges to those inferred from Perturb-seq. We note that steps 4 and 5 are specific to K562, and all performance evaluations were restricted to three traits relating to hematopoeisis; however, this strategy is generalizable to any combination of cell line and corresponding traits that can be identified through heritability enrichment analysis (Ota et al., 2025).

The results in Figure 3 were calculated for a cohort size of 10,000. In Table 1, we report the performance of each of these strategies across multiple cohort sizes, and the last row shows the performance of our final context-aware KGWAS. Our results confirm that these sparsification strategies consistently reduce the number of edges in KGWAS while increasing recall of independent loci identified in full cohort GWAS across a wide range of cohort sizes. Interestingly, leveraging context-specific information, such as gene expression and similarity of transcriptional state after genetic perturbation, leads to an extremely efficient KG with a ∼19-fold reduction in edge count and significantly higher predictive performance for relevant traits.

Table 2 reports baselines relative to the final context-aware KGWAS model to ablate edge contributions: (A) randomize G2G edges, (B) drop G2G edge, and (C) randomize V2G and G2G (the edge counts are slightly higher because no gene prioritization is applied on those baselines, to remove contextual information). Replacing G2G edges with those derived from Perturb-seq outperforms those baselines across all sample sizes, underscoring the benefit of incorporating contextual information. In addition, baseline (B) indicates that sparsified V2G edges alone yield better performance than full V2G edges alone (Fig. 3A), consistent with our hypothesis that the high-confidence V2G subset improving the signal-to-noise ratio. Finally, removing G2G edges performs better than randomizing them, suggesting that corrupted G2G structure is more disruptive than the absence of G2G edges.

Table 1: Top 100 independent loci replicated in large cohort GWAS. All values are summed across three selected traits: MCH, IRF, RDW. The top and bottom groups show models without and with contextual information, respectively.

|  | # Edges | Small cohort sample size | | | | | |
|---|---|---|---|---|---|---|---|
|  |  | 1,000 | 2,500 | 5,000 | 7,500 | 10,000 | 50,000 |
| GWAS | 0 | 14 | 15 | 29 | 36 | 49 | 201 |
| KGWAS | 12.1M | 27.00 (1.00) | 24.67 (1.53) | 37.67 (2.08) | 55.00 (1.73) | 62.33 (2.52) | 221.00 (4.36) |
| + Remove gene programs | 12.1M | 26.33 (1.53) | 24.00 (2.65) | 36.33 (1.53) | 53.33 (2.31) | 65.33 (2.08) | 220.00 (4.36) |
| + Prune edge types | 3.4M | 25.67 (3.06) | 26.33 (2.52) | 46.33 (2.52) | **58.33 (2.52)** | 67.33 (3.51) | 221.00 (2.00) |
| + Collapse GG edges | 2.3M | 25.67 (3.06) | 26.00 (2.00) | 46.33 (0.58) | 55.00 (2.65) | 66.67 (1.53) | 218.67 (2.52) |
| + Prioritize genes | 1.3M | **32.67 (3.79)** | **36.00 (2.65)** | 52.33 (1.53) | 55.00 (1.00) | 78.00 (5.29) | 226.33 (0.58) |
| + Replace with perturb graph | 625K | 30.67 (2.31) | 34.67 (4.04) | **53.00 (2.00)** | **58.33 (2.08)** | **79.67 (2.08)** | **229.00 (3.61)** |

## 5.3 CONTEXT-AWARE KGWAS IMPROVES INTERPRETABILITY OF DISEASE CRITICAL NETWORKS

An important feature of KGWAS is the interpretability provided by the underlying graph attention network, whose attention weights can be interpreted as disease critical networks, connecting variants,

Table 2: Ablation studies on edge contributions to context-aware KGWAS. Top 100 independent loci replicated in large cohort GWAS over MCH, IRF, RDW traits are reported.

| | # Edges | Small cohort sample size | | | | | |
| --- | --- | --- | --- | --- | --- | --- | --- |
| | | 1,000 | 2,500 | 5,000 | 7,500 | 10,000 | 50,000 |
| Context-aware KGWAS (⋆) | 625K | 30.67 (2.31) | 34.67 (4.04) | 53.00 (2.00) | 58.33 (2.08) | 79.67 (2.08) | 229.00 (3.61) |
| ⋆ with randomized G2G | 863K | 24.67 (2.31) | 27.33 (1.53) | 47.33 (2.08) | 54.33 (1.15) | 67.33 (4.73) | 221.33 (3.06) |
| ⋆ with dropped G2G | 747K | 27.67 (2.52) | 29.67 (0.58) | 38.33 (2.31) | 51.00 (1.73) | 72.00 (2.00) | 224.33 (4.73) |
| ⋆ with randomized V2G and G2G | 863K | 19.67 (1.15) | 26.33 (4.93) | 43.00 (12.74) | 50.67 (1.52) | 61.67 (5.69) | 211.00 (3.00) |

genes and programs to specific traits. Based on our above observations, we hypothesized that the large size of the KG in KGWAS, and in particular the existence of multiple redundant paths, may undermine the reproducibility of inferred disease-critical networks, and that restricting the KG to contextually relevant information should improve interpretability.

We present examples of disease critical networks for the rs61759901 variant derived from KGWAS, and context-aware KGWAS in Figure 4. rs61759901 is a significant variant for MCH in full cohort GWAS, but not significant in the 10,000 sub-sampled GWAS. In our experiments repeated with three random seeds, it is promoted by KG-WAS two times, and three times by context-aware KGWAS. For KGWAS, limited V2G and G2G consistency results in many edges appearing only once and includes edges to genes that are not significant in the K562 cell line (shown with a gray border). In contrast, the sparser context-aware KGWAS network results in notably more consistent attention scores and implicates fewer overall genes.

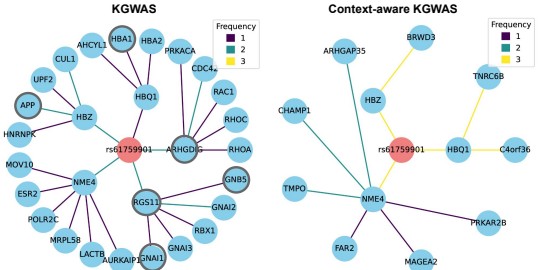

Figure 4: Consistency of disease critical networks for the rs61759901 variant in KGWAS (left) and context-aware KGWAS (right). Each plot aggregate nodes and edges from three seeded models trained on the MCH trait. Genes that are not significant in the K562 cell line (n=6) are shown with a gray border.

Since we trained context-aware KGWAS on blood cell traits using perturbation data from K562 cells, which are derived from a chronic myelogeneous leukemia (CML) patient at blast phase, an advanced stage of CML (Lozzio & Lozzio, 1975), we examined whether relevant biological relationships are recovered by the network attention scores. Hallmarks of CML at blast phase include genomic instability and aberrant mitochondrial metabolism leading to additional DNA damage via increased production of reactive oxygen species (ROS; Senapati & Sasaki, 2022). We identified a disease critical network containing a number of genes that are generally implicated in these processes (Figure 4): regulation of chromosome alignment and segregation (CHAMP1; Li et al., 2026), organization of nuclear envelope and chromatin (TMPO; Vadrot et al., 2023), and mitochondrial ROS generation (NME4; Schlattner et al., 2021). Further genes in this network are implicated as oncogenes (MAGEA2; Weon & Potts, 2015) and in cell signaling and adhesion mediated by small GTPases, which has been linked to CML (ARHGAP35; Thomas et al., 2008; Zhang et al., 2022).

## 6 CONCLUSION

We have presented context-aware KGWAS, demonstrating that cell-type-specific functional genomics can significantly enhance the power and interpretability of GWAS discovery. We show that general-purpose KGs contain substantial redundancy that may dilute biological inference. By pruning low-confidence edges and incorporating direct causal evidence from Perturb-seq, we achieved a 19-fold reduction in graph size while simultaneously improving the recall of independent loci by over 20% in data-scarce regimes. This results suggests that for complex traits with known relevant tissue types, context-specific priors provide a powerful framework for deriving mechanistic insights from GWAS.

A key advantage of our approach is the high consistency of inferred disease critical networks. We note that while higher consistency is partly expected in a sparser graph due to the reduced edge

space, the concurrent improvement in predictive performance confirms that this consistency reflects true biological signal rather than an artifact of sparsity. The increasing availability of Perturb-seq datasets will enable extending this framework to diverse tissues by matching multiple disease traits with a corresponding cell-type-specific Perturb-seq atlas or "virtual cell", moving us closer to a context-driven paradigm for genetic discovery.

## MEANINGFULNESS STATEMENT

Our work trains a graph neural network that represents functional genomics and biological interactions as the latent structure linking GWAS variants to genes and gene-to-gene relationships. We replace a large, generic knowledge graph with a pruned, trait- and cell type-specific graph by incorporating Perturb-seq data. After training on GWAS summary statistics as targets, the model's edge attention identifies regulatory interactions that matter for the relevant traits. Our results show that the approach improves locus discovery in small cohorts and produces more faithful and mechanistically interpretable disease-critical networks, providing deeper and more actionable insight into the mechanisms underlying complex traits.

## ACKNOWLEDGMENTS

The authors acknowledge the use of Gemini (Google) for assistance with improving grammar and wording throughout the manuscript, and we take full accountability for the contents. Some elements in Fig. 1 were created in BioRender.

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
