# OpenReview forum: "Incorporating contextual information into KGWAS for interpretable GWAS discovery"
_ICLR.cc/2026/Workshop/LMRL — ICLR 2026 Workshop LMRL Poster_

### Official Review · Reviewer_jUiX · 2026-02-18
**Graph pruning and context-aware extension of KGWAS using Perturb-seq**

**Rating:** 7
**Confidence:** 3

**Review:**

**Summary**

The paper proposes "context-aware KGWAS", an extension of the existing graph neural network-based KGWAS framework that prunes the knowledge graph and incorporates context-aware gene-gene edges derived from Perturb-seq data. The goal is to improve recovery of loci identified in large-cohort GWAS when applied to small cohorts. The authors show that pruning redundant edges can substantially reduce graph size without degrading KGWAS performance, and that incorporating Perturb-seq-derived G2G edges can further reduce graph size while improving locus recovery performance.

**Strengths**
- Clear motivation for incorporating context-specific information from Perturb-seq, providing useful empirical insights into graph pruning and construction for KGWAS.
- Consistent performance improvements across multiple cohort sizes.
- Reasonable ablations investigating pruning and edge contributions.
- The approach leverages representation learning on knowledge graphs, aligning well with the workshop scope.

**Weaknesses**
- The contribution is primarily an incremental extension of KGWAS, focusing on knowledge graph pruning and adding context-specific edges, rather than introducing a substantially new modeling framework.
- The manuscript describes Perturb-seq derived gene-gene edges as "causal, cell-type-specific". While perturbation data provides causal information about downstream transcriptional effects, connecting genes based on similarity of perturbation response profiles reflects functional similarity rather than direct causal relationships between perturbed genes. This distinction should be clarified.
- Interpretability claims in Fig. 4 rely heavily on context-specific G2G edges derived from Perturb-seq, and the resulting disease critical network may therefore be limited by Perturb-seq coverage. While the context-aware graph appears to improve stability of identified edges across random seeds, the lack of overlap between two-hop neighbors identified by standard KGWAS and the context-aware KGWAS highlights that downstream network interpretation remains strongly dependent on graph construction choices. This tradeoff between context specificity, stability, and graph coverage should be clarified.
- Figure readability could be improved, particularly alignment and legend clarity (e.g., clearer definition of edge "frequency").

---

### Official Review · Reviewer_pMYF · 2026-02-21
**Evaluation of the Context-Specific KGWAS**

**Rating:** 7
**Confidence:** 4

**Review:**

The work is methodologically solid and carefully structured. The evaluation follows the original KGWAS protocol, and multiple cohort sizes (1k–50k) are tested. The ablation studies are thorough, and the manuscript is clearly written and logically organized. This work is incrementally novel but conceptually meaningful. While KGWAS and graph attention–based interpretability are not novel, the systematic sparsification of KGWAS to reduce redundancy in biomedical discovery represents a novel and valuable contribution. The work is practically significant for small-cohort GWAS, with potential applications to rare diseases. The reported reduction in edge count, combined with improved or comparable performance, is practically important because it enables faster training and reduced computational resource requirements.

There are, however, some limitations. The initial setup may be suboptimal. In the Perturb-seq analysis, the cosine similarity distribution for positive gene pairs is bimodal, with a first peak between −0.25 and 0.25 indicating low directional similarity and a much smaller peak between 0.25 and 1.0. Additionally, some claims are phrased as hypotheses rather than directly demonstrated findings. For example, the statement that many V2G edge types potentially contain trans-effects that may be redundant with G2G connections appears speculative. It would also be helpful for the authors to provide more intuition or analysis explaining why removing edge types with few connections and disproportionately many self-edges improves performance. Although the CRISPR screen data used in the Perturb-seq analysis provides a causal perturbation framework, the attention weight–based importance scores derived from the graph attention model do not themselves constitute causal evidence. Clarifying this distinction would strengthen the causality claims. Finally, the project relies solely on the human leukemia-derived cell line K562 using CRISPR interference. It would be valuable to assess the generalizability of the approach across additional cell types and unrelated diseases in future work.

Overall, this is a strong and well-executed project that is well suited for a workshop venue.

---

### Meta-Review · Area_Chair_xxuB · 2026-02-25

**Recommendation:** Accept (Poster)
**Confidence:** 4

**Metareview:**

Accept.

---

### Decision · Program_Chairs · 2026-03-02

**Decision:**

Accept (Spotlight)

**Comment:**

Please see the meta-review.